# Sales of Sugar-Sweetened Beverages in Australia: A Trend Analysis from 1997 to 2018

**DOI:** 10.3390/nu12041016

**Published:** 2020-04-07

**Authors:** William S. Shrapnel, Belinda E. Butcher

**Affiliations:** 1Shrapnel Nutrition Consulting Pty Ltd., 790 Pinnacle Road, Orange, NSW 2800, Australia; 2WriteSource Medical Pty Ltd., Lane Cove, NSW 2066, Australia; bbutcher@writesourcemedical.com.au; 3School of Medical Science, University of New South Wales (UNSW), Sydney, NSW 2052, Australia

**Keywords:** Keywords: sugar-sweetened beverages, soft drinks, sales, trends, added sugar

## Abstract

Lowering intake of sugar-sweetened beverages (SSBs) is being advocated as an obesity prevention strategy in Australia. The purpose of this study was to extend on previous reports of trends in national volume sales of SSBs. Data were extracted from commercially available datasets of beverage sales (AC Nielsen (1997–2011) and IRI Australia (2009–2018)). Linear regression was used to examine trends for the period 1997 to 2018. Per capita attribution of volume sales and sugar contribution was estimated by dividing by the Australian resident population for the relevant year. Per capita volume sales of SSBs fell 27%, from 83L/person to 61L/person, largely driven by declining sales of sugar-sweetened carbonated soft drinks (76 to 45L/person). Volume sales of non-SSB increased, from 48 to 88L/person, the largest contributor being pure unflavoured still waters (6 to 48L/person). Volume sales of non-SSBs have exceeded those of SSBs since 2015. The yearly contribution of SSBs to the sugar content of the national diet declined from 9.0 to 6.4kg/person. Major, long-term shifts are occurring in the market for non-alcoholic, water-based beverages in Australia, notably a fall in per capita volume sales of SSBs and an increase in volume sales of water. Both trends are consistent with obesity prevention strategies.

## 1. Introduction

Since their inception the Australian Dietary Guidelines have recommended limiting the intake of refined or added sugars, the early public health concerns relating primarily to the risk for dental decay and the potential for nutrient dilution [1,2,3]. Although added sugars were initially seen as relatively benign in relation to cardiometabolic risk, recent observational studies have shown associations between intake of sugar-sweetened beverages (SSBs), the major source of added sugars in many western diets, and weight gain, type 2 diabetes, hypertension, cardiovascular disease and all-cause mortality [4,5,6,7,8,9]. Some caution is required in interpreting these associations as high consumption of SSBs tends to cluster with tobacco smoking, low physical activity and poor dietary quality raising the possibility that these drinks serve as a marker for unhealthy lifestyles [8]. Nevertheless, the consistency of the associations between SSBs and weight gain have resulted in a focus on lowering intakes of SSBs as an obesity prevention strategy in the United States, Australia and elsewhere [10,11]. The recommendation for their restriction in national diets is relatively uncontroversial as most SSBs are nutrient-poor with no known health benefits beyond hydration.

Consumption of SSBs is now in decline in several countries. In the United States, Nationwide Food Consumption Surveys and the National Health and Nutritional Examination Surveys (NHANES) have enabled analyses of trends in intake of SSBs over time. After a four-decade climb in the consumption of these drinks, which corresponded with an increase in rates of obesity, intake of sugar-sweetened drinks began to fall in the early 2000s [12]. Using NHANES survey data, between 2003 and 2014 the percentage of US adults and children consuming SSBs on the day of the survey declined, as did the percentage of dietary energy from these beverages [13].

In Australia, a comparison of national dietary survey data collected in 2011–2012 with that from 1995 showed a fall in the percentage of the population consuming SSBs, a fall in the percentage of dietary energy from these drinks, and a fall in the percentage of dietary energy from added sugars in the diets of adults and children [14]. These data are consistent with trend analyses of volume sales of non-alcoholic, water-based beverages in Australia which showed declining sales of SSBs since 1997 [15,16]. The objective of the current study is to update the trend analysis to include the years 2012 to 2018, thereby providing data on volume sales of SSBs and other water-based drinks over a 22-year period.

## 2. Materials and Methods

For the purposes of this study, water-based beverages were defined as beverages that were ready to drink from their packaging and were predominantly water-based. Beverage categories included carbonated soft drinks, energy drinks, sports drinks, mixers, iced teas, sparkling waters and still waters. In contrast to previous studies in this series, fruit juices, sweetened milk-based beverages, such as chocolate milk, and kombucha were also included. Cordial, frozen drinks, syrup-based carbonated soft drinks and tap water were excluded from the analysis as data for these beverages were not available within the datasets used. Beverages were divided into sugar-sweetened and non-sugar-sweetened beverages. Non-sugar-sweetened beverages included all diet varieties, as well as plain and unflavoured mineral waters and still waters. For the purposes of this analysis kombuchas were considered to be non-sugar-sweetened, although not all brands on the Australian market are eligible for a low sugar claim. Alcoholic beverages and pre-mixed alcoholic beverages were excluded from this analysis, and there was no consideration of tea and coffee. This work is an extension of previous Australian research which relied on volume sales data gathered by AC Nielsen [16]. That dataset has been discontinued so a new dataset was sourced from IRI Australia, a market research company servicing the consumer packaged goods industry, and used in conjunction with the AC Nielsen data. The purchasing data sourced from IRI Australia are reflective of annual grocery and convenience store volume sales of water-based beverages nationally and were provided for 2009–2018. The IRI Australia dataset covers the majority of the grocery segment but only 65% of convenience stores nationally, so volumes were adjusted to estimate the full purchasing data in this segment.

To test the compatibility of the two datasets, data from the overlapping years, 2009 to 2011, for identical segments, were mapped. The inflation factor for each segment was determined by taking the AC Nielsen volume and dividing it by the IRI Australia volume for each of the three years individually. An average adjustment factor was determined and then applied across all segments in the IRI Australia dataset, thereby adjusting volume sales to match the previously supplied data from AC Nielsen for the period 1997–2011 (which were used in the first two studies in this series). In this case, the adjustment factor was 1.54. This adjustment factor takes into account the fact that the IRI Australia dataset does not contain information on all convenience volume sales, plus an estimate of foodservice, vending and dining purchases, as per the previous report [16]. It should be noted that all segments reported lower volume sales in the IRI Australia dataset compared to the AC Nielson dataset: for example, before adjustment the total soft drink sales in the AC Nielson dataset in 2009 were 2,025,861 L and 1,296,303 L in the IRI Australia dataset. Following adjustment, the value in the IRI Australia dataset was 1,996,306 L (1,296,303 × 1.54 L). The AC Nielsen dataset did not contain information on fruit juices, sweetened milk-based beverages or kombucha, so these were excluded from this adjustment exercise. Unadjusted data for fruit juices, sugar-sweetened milk-based beverages and kombucha are presented for the period 2009 to 2018.

For the period where the datasets overlapped (2009–2011) the adjusted IRI Australia dataset was used for the 22-year trend analyses, rather than the AC Nielsen dataset. This also enabled a longer assessment of trends in volume sales of juice, juice drinks and sugar-sweetened milk-based beverages, which were only available in the IRI Australia Dataset, and therefore are presented as figures from 2009 to 2018 only.

### Statistical Considerations

Trends in volume sales per annum. Volume sales were examined over the period 1997 to 2018 in per capita terms (litres/person/year). Per capita trends were population adjusted by dividing the annual volume of sales by the estimate of the Australian resident population for that year obtained from the Australian Bureau of Statistics.

The proportion of water-based beverages coming from each beverage category, or volume share, was estimated for each year for the 1997 to 2018 period. The proportion was calculated as the annual volume of sales divided by the total annual volume of sales expressed as a percentage. Linear regression analysis was used to assess trends in volume sales over time, with the beta-coefficient reflecting the change in volume sales per annum. Annual growth rates were calculated as
(1)Annual Growth Rate=(ve−vb)vbn×100,
where *v_e_* is the volume of sales at the end of the period, *v_b_* is the volume of sales at the beginning of the period, and *n* is the number of years in the period, in this case 22.

Trends in sugar contribution from water-based beverages per annum. The sugar contribution to the food supply from water-based beverages was determined by multiplying the annual volume sales by the concentration of sugar per 100 ml for each category of beverage [15].
(2)$$\mathrm{Sugar}\;\mathrm{contribution}\;k=\mathrm{volume}\;\mathrm{sales}\;\left(L\right)\times 10\times \mathrm{sugar}\;\mathrm{concentration}\;\left(\frac{g}{100\mathrm{mL}}\right)\times 10^{-3}$$

The annual growth rate in sugar contribution was determined as for the annual growth rate in volumes, substituting the sugar contribution for the volume of sales. The sugar contribution is presented in per capita terms. 

All analyses were conducted using Stata MP v16 for Mac (College Station, TX, USA).

## 3. Results

### 3.1. Trends in Volume Sales

Over the period 1997 to 2018, the per capita volume sales of non-alcoholic, water-based beverages in Australia increased from 131 L/person to 149 L/ person, a 14% increase (Table 1). Per capita volume sales of SSBs fell from 83 L/person in 1997 to 61 L/person in 2018, a fall of 27%. Over the same period, volume sales of non-SSBs increased from 48 to 88 L/person, an 85% increase driven initially by non-sugar-sweetened carbonated soft drinks and then by bottled water in the latter decade (Figure 1a). As a consequence, the proportion of volume sales of SSBs fell from 64% to 41% of total volume sales of non-alcoholic, water-based beverages. The reduction in volume sales of SSBs was driven largely by falling sales of sugar-sweetened carbonated soft drinks (Figure 1b). Over the period 1997 to 2018, per capita volume sales of sugar-sweetened carbonated soft drinks fell from 76 to 45 L/person, a decline of 41%. In 1997, per capita volume sales of sugar-sweetened soft drinks comprised 58% of all non-alcoholic, water-based beverage sales in Australia but by 2018 this figure had fallen to 30%. In the same 22-year period, per capita volume sales of non-sugar-sweetened carbonated soft drinks increased by 14%, from 23 to 26 L/person, though sales peaked in 2007 and then gradually declined.

Some categories of SSBs, such as energy drinks, sports drinks and iced tea, have shown large annual growth rates over the period of study, albeit from small bases following their introduction in the late 1990s (Figure 2a). Volume sales of these beverages now comprise 9%, 6% and 3%, respectively, of sales of water-based, SSBs. Per capita volume sales of mixers, such as ginger ale and tonic water, increased marginally between 1997 and 2018, the proportion of mixers that were sugar-sweetened falling from 43% to 36%.

The greatest increase in per capita volume sales of any category was in pure water (bottled water), which increased exponentially between 1997 and 2018 (Figure 2b). Volume sales in 1997 were 5.8 L/person and by 2018 this had increased to 47.9 L/person, with sharper rises occurring since 2012. Per capita sales of bottled water in 2018 represented 36% of all sales of non-alcoholic, water-based beverages in Australia, up from 9% in 1997.

Data on per capita volume sales for juice, juice drinks and flavoured milks were only available for the period 2009 to 2018. During this period, volume sales of juice drinks (<100% juice) declined slightly, while sales of 100% juice dropped sharply, by 48% (Figure 3).

Per capita volume sales of flavoured milk grew steadily from 6.1 L/person to 9.5 L/person, an annual growth of 5.64%. Volume sales in this category are only 4% of those of juice/juice drinks. 

Kombucha is a relatively new category in supermarkets and convenience stores with increasing sales.

### 3.2. Trends in Sugar Contribution

Using this market research data for drinks, sugar contribution from sugar-sweetened water-based beverages to the national diet declined from 9.02 kg/person to 6.35 kg/person over the period 1997 to 2018 (Figure 4). Over the same period, per capita sugar contribution from carbonated soft drinks declined from 8.41 kg/person in 1997 to 4.95 kg/person in 2018, or a decrease of 0.17 kg/person per year (Figure 4). These data do not reflect consumption data as no consideration has been given to unconsumed beverages and waste in the current study or to sex and age differences in SSB consumption. 

## 4. Discussion

The 22-year trend analysis reported here indicates that two fundamental shifts are occurring in the market for non-alcoholic, water-based beverages in Australia. Firstly, per capita volume sales of SSBs are in long-term decline, falling 27% between 1997 and 2018. Volume sales of non-sugar-sweetened beverages have exceeded those of their sugar-sweetened counterparts since 2015. The fall in volume sales has been primarily driven by falling sales of sugar-sweetened carbonated soft drinks, which are declining at a rate of 1.9% per year. In turn, the contribution of SSBs to the sugar content of the Australian diet has declined steadily, largely as a result of the fall in volume sales of sugar-sweetened carbonated soft drinks. As a consequence, the sugar contribution of SSBs to the Australian diet would now be expected to be below the figure of 21% observed in the last national dietary survey, conducted in 2011–2012 [17]. Nevertheless, it is likely that these beverages remain the major source of sugar in the Australian diet.

Secondly, per capita volume sales of water have increased exponentially, the annual growth rate of the major category, unflavoured pure waters, being 16%. Water now out-sells sugar-sweetened carbonated soft drinks in Australia by a considerable margin.

Dietary surveys and market research conducted overseas indicate that these two trends are not unique to Australia. The percentage of dietary energy from SSBs and soft drinks has been declining in the United States, in both adults and children, since 2003 [13]. In that country, the per capita volume of soft drinks, sugar-sweetened and non-sugar-sweetened, available for consumption peaked in the late 1990s and has been falling since [18]. After two decades of strong growth, sales of bottled water surpassed those of carbonated soft drinks in the United States in 2016, thereby making water the largest beverage category by volume [19]. Between the two Canadian Community Health Surveys—Nutrition, conducted in 2004 and 2015—substantial falls were observed in consumption of sugar-sweetened soft drinks and juice, and there were sizeable increases in consumption of water, in both sexes and in younger and older adults [20]. The same surveys indicated falls in dietary intake of total sugars and energy in children and adults [20,21]. In the United Kingdom, sales of carbonated soft drinks fell by 8% between 2011 and 2016 while sales of water increased by 35% [22]. Increasing sales of low sugar soft drinks and declining sugar contributions from soft drinks have been observed in the United Kingdom since the announcement of the Soft Drinks Industry Levy in 2016 [23].

Consumer purchases of ready-to-drink beverages have historically been driven by the need for enjoyment and convenience, but other factors are now at play. According to market research firm Euromonitor International, health is the key issue driving changes in the soft drink industry in Australia [24]. Advice to lower intakes of sugar, especially from SSBs, has been promulgated in national dietary guidelines, obesity prevention strategies, health professional advice and associated media coverage for decades and the trends observed in this study suggest the general public is responding. However, it appears that the tight logic of reducing sugar intake to prevent or manage overweight is not the only dimension of health at play in beverage choice. In particular, younger consumers are embracing “wellness”, a more holistic concept of health incorporating physical, emotional, mental, social and spiritual wellbeing, and including concern for the environment and animal welfare [25,26,27,28]. In relation to foods and beverages, the pursuit of wellness is associated with a desire for natural ingredients. Although the definition of the term natural has proved elusive for the Food and Drug Administration in the United States, market research indicates that the general public associates it with simplicity, authenticity and minimal processing [29,30]. Aspartame and other non-nutritive sweeteners appear to fall outside these parameters: 42% of Australian respondents to a market research study stated that “does not contain artificial sweeteners” is a preferred food attribute [31]. Similar negative sentiments have been observed in the United States, despite authoritative reviews by regulatory agencies indicating that non-nutritive sweeteners are safe [32,33].

The interplay of these two concepts of health may explain the trends in volume sales of carbonated soft drinks observed in this study. Between 1997 and 2007, the inexorable decline in volume sales of sugar-sweetened soft drinks was accompanied by only partial replacement with similar beverages containing non-nutritive sweeteners (Table 1). In the subsequent decade, volume sales of non-sugar-sweetened soft drinks levelled out and then fell slightly, despite concerted marketing of these products. Falling volume sales of non-sugar-sweetened carbonated soft drinks is being observed in the United States to an even greater extent [34]. As a result of this combination of trends the proportion of carbonated soft drinks in the overall water-based beverages category has fallen from 75% to 47% over a 22-year period, implying that conventional soft drinks, with and without sugar, are failing to satisfy the changing preferences of health-conscious consumers.

Some contradictory trends were observed in the current study. Volume sales of energy drinks and sports drinks, both of which are primarily sugar-sweetened, have increased which suggests that consumer perceptions of sugar and its associated kilojoules are not universally negative. These beverages are considered to be functional, with caffeine being the active ingredient in energy drinks. Consumers of these products may consider sugar to also play a functional role through the provision of dietary energy or at least to pose no health threat to individuals with active lifestyles. The dimension of health underpinning the popularity of sports and energy drinks is more akin to vitality or high performance than the prudence of sugar reduction or the inner calm of wellness. Despite their growth in volume sales in Australia, sports and energy drinks remain minor components of the sugar-sweetened beverage category and are relatively minor contributors to intakes of added sugars.

Consumer desire for health and wellness may also underpin the recent success of kombucha which combines lower sugar content with natural ingredients and purported functionality associated with fermentation.

An unexpected finding of this study was the substantial fall in per capita volume sales of 100% fruit juice since 2009, the rate of decline being greater than that of sugar-sweetened carbonated soft drinks. As volume sales of fruit juice drinks (<100% juice) over this period were relatively flat, it would not appear that consumers are exchanging pure juice for presumably less expensive juice drinks. Rather, the whole category is in decline and is being led down by 100% juice. Volume sales of flavoured milk, the only other nutrient-rich beverage among those considered in this study, are trending up. However, volume sales of flavoured milk are a small fraction of those of juice/juice drinks and the contribution of flavoured milk to total and added sugar intake in the Australian diet is small [35].

Overall per capita volume sales of water-based beverages gradually increased during the 22 years considered in this study, the annual growth rate being 0.64%. This trend has occurred against a backdrop of declining alcohol consumption in Australia which has fallen to the lowest level since the early 1960s [36]. The degree to which health concerns are driving changes in both sugar and alcohol consumption and the possible interaction between the markets for water-based beverages and alcoholic beverages would be interesting subjects for research.

Strengths and limitations: The current study is based on volume sales data from two market research sources, AC Nielsen Scan Track and IRI Australia, and some discrepancies between the two datasets were evident. For example, “Mineral Water Coloured” is considered as “Soft Drink” in the IRI Australia dataset but is not included as part of the standard “Soft Drink” Category. This may result in the 2009–2018 dataset having less mineral water and more soft drink than the 1997–2008 dataset. In addition, the datasets did not cover the full sales of water-based beverages, and some imputation had to occur to inflate the values to be consistent to include all convenience volume sales, plus an estimate of foodservice, vending and dining purchases. Sugar contribution of beverages was only considered for those categories for which there were 22-year data. However, the exclusion of juice, milk and kombucha drinks from trends in sugar contribution will not have distorted estimates of recent sugar contribution appreciably. In addition, the covered categories do not include all beverage types such as hot drinks (e.g. coffee with syrup) which contribute to the overall SSB intake. Therefore, our estimation of the sugar contribution of SSB in Australia is likely to be an underestimation. As the datasets did not provide consumption information, nor information on which products were purchased together, we are unable to identify whether the overall per capita reduction is masking an increased or maintained risk to those consumers who may not have reduced their intake of SSBs. The major strength of this study is its longitudinal nature with 22 years of data, building on the previous report with 15 years of data. The inclusion of additional beverage categories such as juice, milk and kombucha has enabled a broader consideration of trends in beverage sales in Australia.

## 5. Conclusions

Major, long-term shifts are occurring in the market for non-alcoholic, water-based beverages in Australia, notably a fall in per capita volume sales of SSBs and an increase in volume sales of water. Both trends are consistent with public health nutrition strategies for obesity prevention and suggest that the downward trend in the percentage of dietary energy from added sugars in the Australian diet may be continuing. 

## Figures and Tables

**Figure 1 nutrients-12-01016-f001:**
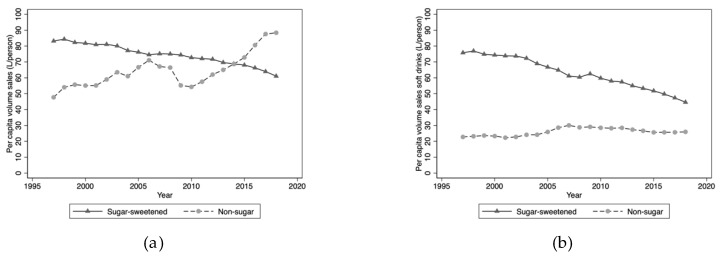
(**a**) Per capita volume sales: Sugar-sweetened water-based beverages and non-sugar water-based beverages; (**b**) Per capita volume sales: Sugar-sweetened carbonated soft drinks and non-sugar carbonated soft drinks.

**Figure 2 nutrients-12-01016-f002:**
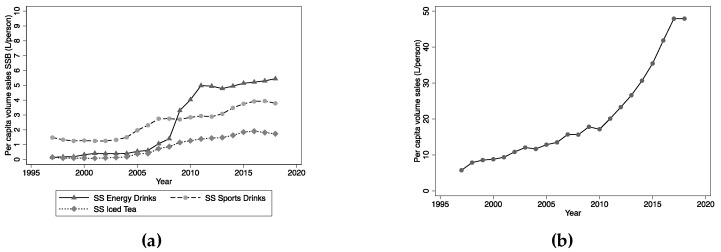
**(a)** Per capita volume sales: Sugar-sweetened sports, energy and iced-tea drinks; **(b)** Per capita volume sales: Pure still water (bottled water).

**Figure 3 nutrients-12-01016-f003:**
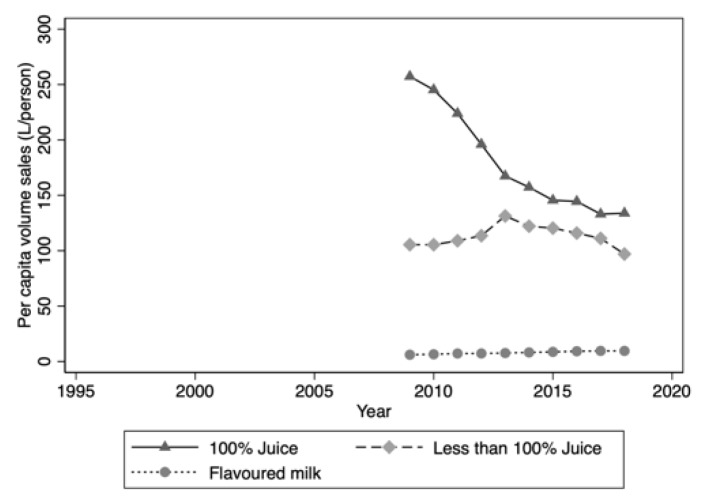
Per capita volume sales: Juice/juice drinks and flavoured milk drinks 2009–2018.

**Figure 4 nutrients-12-01016-f004:**
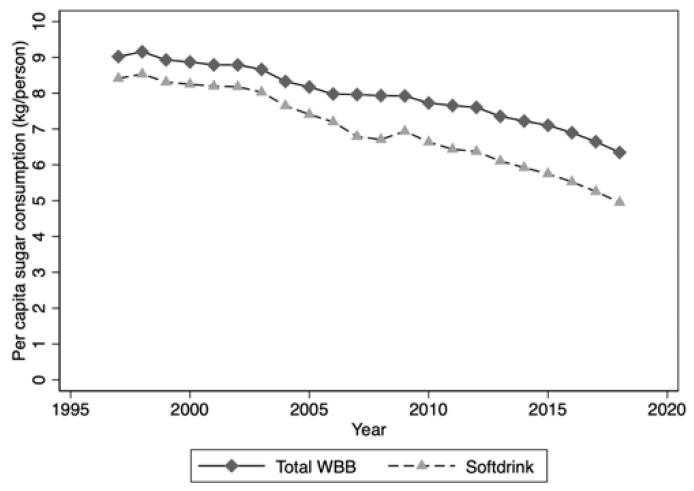
Per capita sugar contribution: Sugar-sweetened water-based beverages and sugar-sweetened carbonated soft drinks. Note: excludes fruit juices and milk.

**Table 1 nutrients-12-01016-t001:** Trends in volume sales of water-based beverages between 1997 and 2018. * Total may not equal sum of sugar-sweetened beverages (SSB) and non-SSB categories due to rounding error.

Beverage Category	Per capita Volume Sales 1997 (L/person)	Per capita Volume Sales 2009 (L/person)	Per capita Volume Sales 2018 (L/person)	Volume Share 1997 (%) *	Volume Share 2018 (%) *	Percentage Change per Capita Volume Sales 2009–2018 (%)	Annual Growth Rate 2009–2018 (%)	Percentage Change per Capita Volume sales 1997–2018 (%)	Annual Growth Rate 1997–2018 (%)
All water-based beverages *									
Total	130.86	129.61	149.42	100.00	100.00	0.67	1.53	14.18	0.64
SSB	83.16	74.40	61.04	63.54	40.85	−0.82	−1.80	−26.59	−1.21
Non-SSB	47.71	55.21	88.38	36.45	59.15	3.27	6.01	85.26	3.88
Carbonated soft drinks									
Total	98.54	91.62	70.58	75.30	47.24	−0.79	−2.30	−28.37	−1.29
SSB	75.78	62.51	44.62	57.91	29.86	−1.69	−2.86	−41.13	−1.87
Non-SSB	22.76	29.12	25.97	17.39	17.37	2.21	−1.08	14.08	0.64
Energy drinks									
Total	0.14	3.56	6.36	0.11	1.33	81.39	7.86	4374.72	198.85
SSB	0.14	3.30	5.44	0.11	1.16	73.61	6.50	3728.39	169.47
Non-SSB	0.00	0.26	0.92	0.00	0.17	NC	25.07	NC	NC
Sports drinks									
Total	1.48	2.79	3.97	1.13	2.66	7.29	4.24	168.25	7.65
SSB	1.48	2.70	3.78	1.13	2.53	7.18	4.03	156.05	7.09
Non-SSB	0.00	0.09	0.19	0.00	0.13	61.48	10.57	6478.99	294.50
Iced tea									
Total	0.15	1.30	1.99	0.16	1.26	44.84	5.30	1217.43	55.34
SSB	0.15	1.15	1.73	0.16	0.78	39.36	5.09	1043.63	47.44
Non-SSB	0.00	0.16	0.26	0	0.56	NC	6.88	NC	NC
Mineral waters									
Total	5.00	4.49 ^†^	9.02 ^†^	3.82	6.03 ^†^	−0.58	10.08	80.47	3.66
SSB	2.17	1.49	1.88	1.66	1.26	−2.27	2.61	−13.47	−0.61
Non-SSB	2.82	0.80	1.17	2.61	2.01	−6.02	4.51	−58.65	−2.67
Mixers									
Total	7.97	7.24	8.36	6.09	5.59	−0.07	1.55	4.89	0.22
SSB	3.41	2.48	3.01	2.61	2.01	9.14	2.11	−11.88	−0.54
Non-SSB	4.55	4.75	5.35	3.48	3.58	−6.98	1.25	17.47	0.79
Unflavoured pure waters									
Total	11.81	20.02	53.88	9.02	36.06	4.32	16.91	356.33	16.20
Still	5.76	17.83	47.91	4.41	32.57	14.32	16.88	732.37	33.29
Sparkling	-	2.19	5.97	-	3.99	NC	17.21	NC	NC
Flavoured Milks									
Total	-	6.06	9.48	-	-	56.42	5.64	NC	NC
Juice									
Total	-	362.59	230.75	-	-	−36.36	−3.64	NC	NC
100% Juice	-	257.21	133.83	-	-	−47.97	−4.80	NC	NC
Less than 100% Juice	-	105.38	96.93	-	-	−8.03	−0.80	NC	NC
Kombucha									
Total	-	0.00	0.47	-	-	NC	NC	NC	NC

NC = not calculable; ^†^ Note, in the IRI dataset (2009–2018), coloured mineral water was considered a soft drink, but not included in the standard soft drinks category. Hence the sum of sugar-sweetened and non-sugar not summing to the total in this category. * Excludes mil, juice, kombucha.

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
