# Peer review of "Sales of Sugar-Sweetened Beverages in Australia: A Trend Analysis from 1997 to 2018"

_nutrients, 2020, doi:10.3390/nu12041016_

Round 1

Reviewer 1 Report

This is an interesting analysis of data. While the authors have used an approach that tests the compatibility of two datasets arising from 2 years of overlapping data it is not clear that these datasets collect the same information. Authors have 'corrected' one dataset in order to align the data but I question whether this is the most appropriate approach given the datasets are likely/appear to have different underlying basis. it is unclear whether the corrections were all in the same direction/magnitude and given that there is a 35% correction for other outlets the authors should provide more information on the completeness of collection, the similarity of collection, the impact of other confounds eg socioeconomic groups, geography, type of store, main shopper etc (all things that I would imagine the datasets contain) in order to convince the reader and be clear that the methodology is appropriate. It may still be more appropriate to present the different trend lines rather than correcting one dataset for the other. I am not an expert on the Australian beverage situation but if sales of hot drinks with sugary additives (eg coffee with syrups) is as large as it is in the US or the UK then the data is likely an underestimation of the situation in Australia. Given the concerns above about the dataset coverage this is likely to expand the underestimation and this is not clearly recognised by the authors within the text.

I would usually expect analyses to consider not only per capita but of consumers - this is especially important in this situation where there is a reduction in SSB and increase in water. Otherwise it is unclear whether the overall per capita reduction is masking an increased or maintained risk to consumers who may not have reduced their intake of SSBs. This issue needs to at least have some comment in the discussion. I do recognise that the dataset does not contain consumption data but I assume it has the potential to assess what was purchased together which may give some information - it may also not be appropriate to make such assumptions if purchases are volumes which could be shared.

Given the extent of work in the UK it is surprising that the only reference to trends in sugar intake in the UK is a single reference to an industry report rather than the more extensive work published by the UK government (ie PHE's evidence and evaluation reports).

Authors should do a check of internal referencing. Also unclear that data refereed to in line 214 is actually clearly seen in table 1.

the title of the article is about SSBs so it is less clear why other aspects of consumption focus so highly - when they are not introduced within the text more thoroughly. While these may be important considerations they go further than a trend analysis of SSBs. For example, it is unclear why the authors talk about intake of vitamins and minerals. While beverages may provide 20% of vitamin C intakes the authors fail to show that this is a nutrient of concern across the diet - unless it is this is an irrelevance and could be construed as promoting an unnecessary health or nutrient claim. in similar vein, the authors comment on the sugar and caffeine content of energy drinks but do not comment about the energy contribution. This could be of considerable public health concern for consumers, the impact diluted by using only per capita assessments. I do think these are public health issues of interest but recommend the authors consider either the title of the article or how they incorporate these aspects more clearly and appropriately within the introduction and discussion.

While I do not have concerns about the unrestricted grant from the beverage council and am not suggesting that the authors are introducing an industry bias, given the issues raised above the authors may wish to review how the discussion/conclusions might be perceived given this grant. the data represents an analysis (subject to comments above) about SSBs but some may argue potential for a bias in the wider text.

Reviewer 2 Report

The authors present an interesting analysis of trends in soft drink and other beverage consumption over the past 22 years. Although such data has some inherent flaws, these were covered nicely in the manuscript. Such strong declines in SSB consumption (or at least sales) is fascinating.

Author Response

We thank the reviewer for their comments.

Reviewer 3 Report

The manuscript by Shrapnel et al. analyses the trends in national volume sales of sugar-sweetened beverages (SSBs). The authors observe that major, long-term shifts are occurring in the market for non-alcoholic, water-based beverages in Australia, notably a fall in per capita volume sales of SSBs and an increase in volume sales of water. Both trends are consistent with obesity prevention strategies. The study is comprehensive and generally supports the authors' conclusions. These findings may benefit from some additional clarification, as detailed below.

Comments

- The authors analyze 22-year trend. Figures 1 and 2 display a different trend in 1995-2005 vs 2005-2015 years. These data should be better explained and studied considering the peculiar socio-economic condition.

- Is there a correlation between trends in volume sales of SSBs and national dietary habits?      

- The impact of the study should be better clarified and detailed.

- Lines 115-116 please insert reference.

- The manuscript should be edited to correct contextual and layout errors.

Round 2

Reviewer 3 Report

No comments